# Rerouting Artificial Light for Efficient Crops Production: A Review of Lighting Strategy in PFALs

Xinying Liu [1,2], Yaliang Xu [1,2,*], Yu Wang [1,2], Qichang Yang [1,2] and Qingming Li [1,2]

[1] Institute of Urban Agriculture, Chinese Academy of Agricultural Sciences, Chengdu 610213, China; liuxinyingya@163.com (X.L.); wangyu05@caas.cn (Y.W.); yangqichang@caas.cn (Q.Y.); liqingming@caas.cn (Q.L.)
[2] National Chengdu Agricultural Science and Technology Center, Chengdu 610213, China
[*] Correspondence: xuyaliang@caas.cn

**Abstract:** A plant factory with artificial light (PFAL) is defined as an advanced agricultural production system with a precisely controlled environment, playing an important role in vertical farming and urban food supply. Artificial light is one of the core technologies in PFALs and accounts for a large part of energy consumption; elevating the light utilization efficiency of plants is vital for the sustainable development of PFALs. Meanwhile, the enclosed structure of the plant factory resulted in the independence of its light environment, indicating that the light environment in PFALs can be custom-made. Lighting strategy is an attempt to reprogram the light environmental parameters in unconventional ways, resulting in innovative lighting modes for energy-saving, high-yield, and high-quality production in PFALs. This article reviewed the recent endeavors aiming to increase light conversion efficiency and nutritive properties of crops by different lighting strategies, providing economic irradiation patterns or modes for various PFALs production goals.

**Keywords:** plant factory; artificial light; lighting strategy; energy conversion efficiency

## 1. Introduction

Plant factories are a type of closed production system, which integrated modern industrial technologies for the annual efficient production of crops [1,2]. As an artificially controlled environment production system, environmental factors in plant factories, such as temperature, humidity, light, $CO_2$, nutrients, and water, are precisely controlled by artificial intelligence systems and electronic sensors, according to the requirement of crops, largely avoiding the restriction of a natural fluctuant environment and contributing to green manufacture [1,3,4]. Consequently, food production in plant factories is not limited by geographical factors, seasonal changes, and available spaces, producing green crops all year round and representing future agriculture [5]. With the continued expansion of the human population and the acceleration of urbanization, plant factories with artificial lighting (PFALs) possess extensive application prospects in the urban life of modern society [6]. Especially in a post-coronavirus era, PFALs provide an alternative path for industrial food manufacture and are also confronted with new challenges.

Light is one of the vital environmental factors in a plant factory and accounts for a large part of energy consumption [7]. For lowering light energy consumption in PFALs, a mass of attempts adopting innovative lighting devices or methods have been developed and tested, aiming to increase the light-use efficiency of plants. As a new-generation light source with outstanding engineering advantages, light-emitting diodes (LEDs) displayed huge potential for indoor crops cultivation [8,9]. Producing LED chips with high wall-plug efficiency (WPE) and long lifespan was important for efficient production in PFALs [10]. Moreover, how to create an ideal light environment meeting the requirement of plants is equally important. The light recipe has been demonstrated to be an efficient method, which increases the energy conversion efficiency of plants by integrating different spectrums with

reasonable ratios [11–13]. Red and blue light usually served as a skeleton spectrum in the spectral recipe, as they are widely acknowledged as photosynthetically active irradiation [14]. Ultraviolet (UV), green, and far-red were treated more as signal spectrums to adjust the morphological traits, biological activity, and nutritive quality of crops [15–17]. Besides, zoom lenses [18] and moveable LED platforms [19] were also shown to achieve impressive effects in energy saving by optimizing light distribution.

During the traditional cultivation of crops, sunlight serves as the main energy resource, and the light intercepted by plants under a natural environment fluctuates and is complicated. In a plant factory, however, the light environment is insusceptible to natural conditions and can be artificially created. Especially with the rise of light-emitting diodes (LEDs) and the developing of artificial intelligence, the light environment in plant factories can be custom-made from different dimensions [20–24]. Consequently, plenty of lighting modes that are quite different from the conventional ones were designed and their practical effects on crops have been investigated, such as intermittent lighting [25], alternate lighting [26], and continuous lighting [27]. Therefore, lighting strategy is an effort to recruit artificial light unconventionally, resulting in innovative lighting mode for energy-saving, high-yielding, and high-quality production in PFALs. This article reviewed the recent practices devoted to economic production in PFALs with unusual lighting strategies; their practical effects and underlying mechanisms were simultaneously introduced, aiming to provide more efficient and energy-saving cultivation methods for varied PFALs production goals.

## 2. Alternate Lighting

Distinct monochromatic spectrum exerts different specific effects on the photomorphogenesis, physiological performance, and developmental status of plants [28,29]. In the natural environment and conventional PFALs cultivation, plants were constantly irradiated by multichromatic sunlight or tailored spectral composition with a fixed ratio, so the potential of the light spectrum on crop production was still underestimated. Alternate lighting is an attempt that alternately irradiates plants with different light spectrums or spectral compositions for efficient crop production, including full-alternated lighting mode and overlay-alternated lighting mode [30–32]. Compared with full-alternated lighting, overlay-alternated lighting was relatively complicated and extendable, which shifted the irradiation time of blue/red light forward/backward based on concurrent lighting, ensuring crops were irradiated by the monochromatic spectrum for a certain time interval [31,33]. The current study showed that some biological characteristics of crops were positively affected when exposed to alternate lighting; meanwhile, with the optimization of parameters, such as spectral composition, alternating intervals, and light intensity, the energy conversion efficiency in PFALs was elevated, and resulted in increased biomass and nutritional quality [26,30].

Blue light and red light were the most frequently used spectrums in alternate lighting, as they were photosynthetically active spectrums, being widely used in PFALs [14,34]. Besides, the physiological effects of both spectrums were extensively reported [7]. Although monochromatic blue and red light failed to meet the requirement of plants and led to dysfunctional photosynthetic responses [35], alternate lighting might be a good resolution. Chen et al. [30] alternately irradiated lettuce with monochromatic red and blue light under plant factory conditions, and the results revealed that phenotype and nutritive properties of lettuce were largely affected; specifically, significantly higher biomass was achieved compared with concurrent irradiation without extra energy consumption [30]. Similar results were also achieved in several other lettuce cultivars, where the biomass was significantly elevated by more than 60% [26]. Alternate irradiation was also a potential strategy for improving the nutritive properties of crops [30,36]. In lettuce, alternate lighting of blue and red light resulted in increased ascorbic acid and decreased nitrate [30]. Pepper exposed to alternate irradiation of two dichromatic lights with a high/low ratio of blue spectrum tended to accumulate anthocyanins and carotenoids, whereas the normal growth was not

affected [36]. Besides, alternate lighting of blue and red-orange light was beneficial for biomass production and lipid biosynthesis in green algae; particularly, lipid concentration was much higher when the last alternation was red-orange light [37].

Surprisingly, alternate lighting might be a potential method of avoiding the negative effects of continuous lighting. A long photoperiod was negative for the normal growth of crops, as it usually caused interveinal chlorosis consequently [38]; nevertheless, continuous 24 h supplemental lighting with alternate blue and red light could alleviate the injury and led to increased net carbon exchange rates in tomato [39]. For lettuce cultivation in a plant factory, alternate lighting of blue and red light over a 24 h photoperiod resulted in significantly increased biomass, while ideal nutritive quality was simultaneously obtained [40].

From above, alternate lighting is characteristic of creating a light environment where crops are alternately irradiated by different spectral compositions. Molecular mechanisms underlying the regulation of alternate lighting on growth still remained to be discovered, whereas it is widely reported that the stimulating effects of alternate lighting are attributed to positive effects on morphology, metabolism, and photosynthesis efficiency [40,41]. Lettuce cultivated under optimal alternating blue and red light treatment tends to have a higher PA/LA (projected area/leaf area) index; meanwhile, photosynthetic efficiency was significantly enhanced [41]. Besides, different monochromatic lights or spectral compositions possess varied physiological or phytochemical effects [42]. For instance, blue light is favorable for the biosynthesis of secondary metabolites and protein, while red light promotes photosynthetic efficiency and photosynthates production. Therefore, alternate lighting could take full advantage of spectrums with different physiological effects, resulting in double benefits of increased growth and enhanced nutritive quality. Meanwhile, the negative effects of constant lighting were simultaneously weakened. For example, excess accumulation of photosynthates caused by constant red light was harmful to photosystem II, while alternate lighting of blue light could activate the physiological processes that consumed them and are used in biomass production. On the other hand, plants sensing different light spectrums and the corresponding photomorphogenesis responses were mediated by specific photoreceptors and signal transduction pathways [30], whereas, different light receptors exert synergistic or antagonistic interaction patterns on the regulation of specific physiological processes depending on the light environment [30,43]. For example, Cry1 and PhyB are synergistic under short photoperiods, while turning into independent and additive when exposed to continuous lighting [43]. Therefore, alternate lighting may erase this discrepancy between different light receptors and lead to efficient growth.

Nevertheless, the reported effects of alternate lighting were controversial [30,40]. This phenomenon was considered to be related to the cultivation stage [40], and design parameters also matter. For full-time alternate lighting, inappropriate alternating intervals exert no stimulating effects on biomass and resulted in remarkable different effects on biomass and nutritive properties [26,30,44]. In general, alternate lighting is an efficient strategy to take full advantage of different spectral compositions during crop production in PFALs, it also might be an innovative platform to reveal the interaction relationship of corresponding light receptors under varied light environments.

## 3. Intermittent Lighting

The rotation of Earth on its axis leads to the alternation of day and night, serving as an important environmental signal for the metabolism and development of plants [45]. In PFALs, the light–dark cycle was independent of sunrise and sunset, providing huge space for innovative regulation of the light environment. Intermittent lighting was another attempt to artificially remold the light environment in PFALs based on photoperiods, which separate the standard 24 h day/night alteration into short light/dark cycles and irradiate plants temporarily [25]. Besides, the short flash created by pulsed LEDs was considered as another kind of intermittent lighting, providing light on the seconds scale, even on the milliseconds scale [46,47]. Up to now, many endeavors have demonstrated that intermittent lighting is beneficial for crop production and possesses large potential

in reducing energy consumption in PFALs [46,48]. Moreover, frequency and duty ratio (duration ratio of light in a whole light/dark cycle) are important design parameters of intermittent lighting [49,50].

Cheng et al. [25] divided the light/dark cycle of 16 h/8 h into short combinations of 8 h/4 h, 6 h/3 h, 4 h/2 h, 3 h/1.5 h, and 2 h/1 h under identical energy consumption, with results revealing that lettuce exposed to intermittent lighting treatments almost all show increased biomass, except for 6 h/3 h. Besides, the sweetness and crispness of lettuce were also influenced [25]. Avgoustaki et al. [48] innovatively integrated intermittent lighting into the dark period of 14 h/10 h light/dark cycle, and the biomass of *Ocimum basilicum* was found to increase by 47%. It is worth noting that, due to the synchronization of lighting time with the fluctuation of electricity price, the energy consumption was simultaneously decreased by 15.9% [48]. Except for light/dark frequency, the light spectrum employed during intermittent lighting was also vital for the final effects. Sweet potato plantlets cultivated in intermittent blue and blue–red light tend to have a higher net photosynthetic rate and leaf area compared with intermittent red light, resulting in significantly higher dry matter [47]. On the other hand, intermittent lighting was capable of enhancing the economic value of crops by inducing the accumulation of secondary metabolites, such as anthocyanin in pak choi and tatsoi [51], or soluble polysaccharides biosynthesis in *Dendrobium officinale* [52].

The stimulating effect of intermittent lighting on growth is partially attributed to its positive regulation of photosynthesis efficiency [47]. Further study revealed that adequate frequencies could improve fluorescence emission parameters of chlorophyll as $Fv'/Fm'$(maximum efficiency of PSII), NPQ (nonphotochemical quenching), ΦPSII (quantum efficiency of photosystem II), ETR (electron transport rate), and ø $CO_2$ (quantum yield of $CO_2$ assimilation) [53]. Notably, intermittent lighting was an effective strategy for regulating the photosynthetic pathway of CAM plants [54]. A short light/dark cycle of 2 h/2 h was reported to switch the photosynthetic pathway of *Dendrobium officinale* from CAM to C3, resulting in increased net $CO_2$ exchange amounts and higher biomass [52]. However, this process is incompletely reversible, as *D. offcinale* maintained an increased net $CO_2$ exchange amount when transferred back to 12 h/12 h [54]. Besides, it was assumed that more photoreceptor cells were formed when exposed to intermittent radiation [47].

Intermittent lighting shortens the periodic light time for plant growth and brings forward dark, indicating that the light energy intercepted by plants in every cycle was decreased. Therefore, the most remarkable influence of intermittent lighting was the metabolic alteration of sucrose [25] and starch [47,55], implying that the regulation of carbohydrate metabolism might be the core mechanism leading to efficient production. Specifically, starch serves as an important intermediate during the coordination of the circadian clock between the growth and environmental adaption of plants, as the metabolism pattern is vital for plant productivity [56–58]. The growth and metabolism of plants during night time majorly depend on starch stored in the daytime [59]; early exhaustion or incomplete consumption of starch at night will result in slow growth [60]. Besides, both starch biosynthesis and degradation are energy-consuming biological processes [59]. Compared with the formation of sucrose in the day time, stored starch transformed to into sucrose at night expends an extra 5/3 ATP [61]. Moreover, the pattern of starch metabolism is affected by light duration time. Kölling et al. [60] found that in Arabidopsis, starch accounts for 14.1% of photosynthates at the early stage of the photoperiod and increased to 32.8% in 6 h, while 43.9% of photosynthates allocated to sucrose at the beginning and afterward decreased to 20%, indicating that starch accumulates with the prolonging of lighting [60]. On the other hand, dark is beneficial for the sustainable growth of plants, as it can alleviate the peroxide damage to the photosynthesis system caused by continuous lighting [62], and also help to maintain the homeostasis of endogenous hormones [25]. Therefore, an ideal light/dark cycle could largely relieve the negative biological effects and energy-wasting process caused by starch over-accumulation, simultaneously ensuring that adequate substrates are provided to meet the requirement of the plant at night.

Although pulsed LEDs could release a short flash, which, theoretically, resembles the pattern of intermittent lighting, their biological effects on plants and energy-saving mechanisms might be quite different [50,63]. Pulsed LED was extensively used in fundamental research to uncover the operation pattern of the photosystem [64]. Optimal combination of frequency and duty ratio could elevate the average net photosynthetic rate of plants, as even the average PPFD remained unchanged [63]. Potato plantlets obtained the highest biomass under the combination of 720 Hz (1.4 ms) and 50% duty ratio, while 180 Hz (5.5 ms) and 50% duty ratio were more economic [49]. Wheat exposed to a light/dark cycle on the milliseconds scale displayed increased photosynthetic rates and decreased lignin, and the biomass achieved was comparable with that under continuous lighting; nevertheless, the light/dark cycle of 0.8 ms/0.2 ms was more energy-saving than continuous lighting [46]. Similar results were also obtained in lettuce, indicating that pulsed light might be more efficient in biomass production in PFALs compared with intermittent lighting on the hour scale, as it, indeed, reduced energy consumption [50].

Notably, this finding might be a great revolution for crop production in PFALs, as the growth process of crops could partially shift to proceed at night, taking into account the diurnal fluctuation of electricity cost, production cost in PFALs will be largely decreased. Moreover, related research may help to further understand the biochemical nature of the photosystem process and the molecular mechanism of how the circadian clock synchronized the growth and environmental adaption of plants by regulating the metabolism rhythm.

## 4. Continuous Lighting

As the sole irradiation source in PFALs, artificial light provides energy and acts as an important environmental signal for the growth and development of cultivated crops. Continuous lighting (CL) is an efficient cultivation strategy by extending the duration of light, theoretically driving crops to grow unceasingly in PFALs [65,66]. Crops exposed to a 24 h photoperiod displayed distinct statuses, as some of them experienced severe physiological damage, while some of them adapted well and obtained increased yield [67]. Particularly, for a life support system in space or basements in the polar regions, the food supply problems could be well resolved by this strategy. Besides, continuous lighting was also supposed to be used in breeding research, as the vegetative growth was stimulated and shortened the time for crop selection.

Continuous lighting has the potential to induce injury to photosynthetic organs, leading to various symptoms, such as photo-oxidative damage, early senescence, and/or decreased photosynthetic efficiency, which are usually mediated by hyperaccumulation of carbohydrates or negative effects on photoreceptors [66]. Velez-Ramirez et al. [66] reported that lettuce achieved increased biomass under CL, while tomato displayed CL injuries. This phenomenon is partly due to differences in CL tolerance. On the other hand, endogenous circadian rhythm may also play an important role. Besides, the phenotype of continuous lighting-induced injury was influenced not only by irradiation intensity, but the light spectrum also exerted a huge influence. Tomato seedlings exposed to continuous red or blue light displayed varied symptoms, while their combination could largely relieve it [68]. On the other hand, photoinhibition caused by continuous lighting is reversible when the duration time is short, while the accumulation of ROS and lipid peroxidation is inevitable in long-term treatment [69]. Integrating continuous lighting with alternate lighting of red and blue light might be a good resolution, as photosynthetic parameters under continuous lighting were similar to that of a 12 h photoperiod; moreover, diurnal metabolism of carbohydrate was also observed [40]. On the other hand, continuous lighting decreased the mineral contents of lettuce, as the growth was largely enhanced. However, it has been reported that an adequate combination of light quality and intensity was able to elevate biomass and mineral content simultaneously [65].

Considering the negative effects of continuous lighting in the long term and disproportionate input–output ratio, it was commonly used in the short term as a pre-harvest strategy for the enhanced nutritional quality of crops, and the harvested biomass was

simultaneously enhanced [15,70]. Nitrate is the main harmful substance that existed in leafy vegetables cultivated in hydroponic conditions, and it was reported that short-term continuous lighting could significantly inhibit the accumulation of nitrate, and this effect was affected by spectral composition and light intensity [15,71]. Red and blue are the generally used spectrums during continuous lighting; moreover, green light was recently demonstrated to be an effective spectrum in reducing nitrate content by elevating the expression abundances of genes coding NR (nitrate reductase) and NiR (nitrite reductase), so activities of the enzyme involved in nitrate assimilation was enhanced simultaneously [72]. Moreover, green light supplementation also contributed to higher photosynthetic rates and antioxidant activity, which partially neutralized the negative effect caused by continuous lighting [73]. The effects of continuous lighting also depend on the light intensity employed. Zhou et al. (2012) found that the nitrate content of lettuce decreased with the increase in irradiation strength, and existing border effect [71]. On the other hand, a prolonged duration of continuous lighting will not further decrease nitrate content, which usually tends to be steady at a specific timepoint.

Soluble sugars and Vc (Vitamin C) are also important quality traits of crops, which could be substantially improved by continuous lighting [27,69–71]. As the direct product of photosynthesis, soluble sugar biosynthesis is significantly elevated by extended lighting, leading to a more delicious taste in crops [72]. Vc, also named as ascorbic acid, exists widely in plants, as an antioxidant chemical, and is beneficial for human health [74,75]. Accumulation of Vitamin C in plants commonly acts as a defense mechanism, helping to wipe off the elevated excitation energy, and maintain the normal operation of photosynthetic apparatus from photooxidative damage under continuous lighting [69]. Besides, the content of Vc and the activities of enzymes involved in Vc biosynthesis were positively correlated with the light intensity [69]. As a result, stronger irradiation usually leads to a higher content of Vc in plants [71]. Besides, phytochemicals with antioxidant ability, such as glutathione [69] and phenolic substances [15], were induced to be increased, and antioxidant activity of the treated plants was also detected to be enhanced [73,76].

## 5. Conclusions

As an important environment signal and sole energy source, light is the essential environmental element during the morphogenesis, growth, and quality formation of crops cultivated in PFALS. The attempts reviewed in this article proved that unconventional lighting strategies, such as alternating lighting, intermittent lighting, and continuous lighting, et al., have great application prospects in PFALs for economical and efficient vegetable production. These findings remind us that the light environment designing of PFALs should ignore regular photobiology knowledge. Meanwhile, based on the functional principles of the above lighting strategies, plentiful lighting modes or methods can be designed and tested for their physiological effects on crop production in PFALs. On the other hand, these lighting modes can also serve as innovative experimental platforms to investigate the regulation mechanisms of light on morphology, physiology, and development of plants.

**Author Contributions:** Conceptualization, Y.X. and Q.L.; investigation, X.L. and Q.Y.; resources, Y.X.; writing—original draft preparation, X.L., Y.W. and Y.X.; writing—review and editing, X.L. and Y.X.; visualization, X.L. and Y.X.; supervision, Q.Y. and Q.L.; funding acquisition, Q.Y. and Q.L. All authors have read and agreed to the published version of the manuscript.

**Funding:** This work was financially supported by the National Key Research and Development Program (No. 2020YFB0407902), Central Public-interest Scientific Institution Basal Research Fund of China (No. 34-IUA-03), and the Local Financial Funds of National Agricultural Science and Technology Center, Chengdu, China (No. NASC2020AR10; No. NASC2021PC03).

**Institutional Review Board Statement:** Not applicable.

**Informed Consent Statement:** Not applicable.

**Conflicts of Interest:** The authors declare no conflict of interest.

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
