# Peer review of "Rerouting Artificial Light for Efficient Crops Production: A Review of Lighting Strategy in PFALs"

_agronomy, doi:10.3390/agronomy12051021_

Round 1
Reviewer 1 Report
Article review: Xinying Liu, Yaliang Xu, Yu Wang, Qichang Yang, Qingming Li,
Rerouting Artificial Light for Efficient Crops Production: A Re-2 view of Lighting Strategy in PFALs
The authors' review discusses the results of a number of recent works carried out in recent years and concerning the cultivation of plants under alternating, intermittent and continuous LED irradiation of various spectral compositions.
The analysis of experimental data by the authors showed that unconventional
lighting strategies have great prospects for application in plant light culture for economical and efficient vegetable production.
The results also show the need for further study, development and design of modes and methods of lighting plants with LED irradiation sources for their physiological effects on crop production. An important conclusion is that lighting modes can serve as innovative experimental platforms for studying the mechanisms of light regulation on the morphology, physiology and development of plants.
Considering all these circumstances, the presented article is recommended for acceptance in the journal of Agronomy
Author Response
Dear reviewer, thank you for your agreement on the current manuscript.
Reviewer 2 Report
Very interesting review article. I think it is very important to discuss lighting strategy in PFALs.
I want to confirm only one point to the authors. "PFALs" were used in some lines, but "plant factory" were also used in the other lines. I think the authors used "PFALs" only when the facility uses no sunlight, and "plant factory" contains greenhouses and PFALs. Is that right? If it is right, this article can be accepted without any revision. If not, please distinguish the words clearly.
Author Response
Dear reviewer, thank you for your agreement on the current manuscript. Besides, it is right that we used "PFALs" for plant factory using artificial light, while "plant factory" contains greenhouses and PFALs.
Reviewer 3 Report
The English must be improved as the manuscript in its current state is practically un-readable.
Some other comments are below.
Line 13: please use a semi-colon (;) instead of a comma (,)
Abstract: The English is quite poor. I suggest this be improved.
Line 24: Change to Plant factories are a type of…
Line 35: era not ear.
Line 61: remove et al.
Line 69-73: This is a run-on sentence.
Line 90: Not the proper reference format
Line 115: remove et al.
Line 155: Not the proper reference format.
Line 159: Not the proper reference format.
Line 195: Not the proper reference format.
Line 230-232: This is not the case for all crops. Tomatoes show CL-injury, but lettuce and cucumbers don’t. Furthermore, peppers show decreased yield under some types of CL but not others. These should be discussed.
Author Response
Dear Reviewer
Thank you for your careful and rigorous review of the current work. Your comments and suggestions are really helpful for us to improve the quality of this manuscript and meet the requirements of Agronomy. Please find the attached file, we have made a point-by-point response to your comments, you can also find the marked changes in the revised manuscript.
